# The Reconstruction of Various Complex Full-Thickness Skin Defects with a Biodegradable Temporising Matrix: A Case Series

**DOI:** 10.3390/ebj6020024

**Published:** 2025-05-14

**Authors:** Julie van Durme, Thibaut Dhont, Ignace De Decker, Michiel Van Waeyenberghe, Kimberly De Mey, Henk Hoeksema, Jozef Verbelen, Petra De Coninck, Nathalie A. Roche, Phillip Blondeel, Stan Monstrey, Karel E. Y. Claes

**Affiliations:** 1Burn Center, Ghent University Hospital, C. Heymanslaan 10, 9000 Ghent, Belgium; 2Department of Plastic Surgery, Ghent University Hospital, C. Heymanslaan 10, 9000 Ghent, Belgium; 3Faculty of Medicine and Health Sciences, Ghent University, C. Heymanslaan 10, 9000 Ghent, Belgium

**Keywords:** wound reconstruction, full-thickness skin defect, biodegradable temporising matrix, synthetic dermal substitute

## Abstract

Background and Objectives: Traditionally, full-thickness skin defects (FTSDs) are covered with split-thickness skin grafts (STSGs). This usually provides an epidermal coverage but entails a high risk of hypertrophic scarring mainly due to the absence of the dermal layer. The Novosorb^®^ Biodegradable Temporising Matrix (BTM) is a novel synthetic dermal substitute that has been used for the reconstruction of various complex and/or large defects in our center. The aim of this article is to evaluate the clinical performance of the BTM as a synthetic dermal substitute for complex FTSD reconstruction in a European context. Materials and methods: This case series focused on the treatment of complex FTSDs with the BTM. After wound debridement, the BTM was applied according to a defined protocol. Once adequate vascularization was observed, the sealing membrane was removed and the neo-dermis was covered with STSGs. Patient demographics, comorbidities, wound defect localization and etiology, wound bed preparations, time of BTM application and removal, time to complete wound healing after STSG, complications, and HTS formation were recorded. Results: The BTM was used to treat FTSDs in six patients with complex wounds from degloving (3), burns (1), ulcerations (1), and necrotizing fasciitis (1). Successful integration occurred in five cases (83%), with one partial integration. The BTM remained in situ for an average of 20.7 days before delamination and STSG coverage. No major complications occurred, though one case had hypergranulation with secondary STSG infection. Two patients were lost to follow-up, while the remaining four had excellent aesthetic and functional outcomes with good-quality scars. Conclusions: Within the limits of this small and heterogeneous case series, the BTM appears to be a promising option for the reconstruction of complex FTSDs of varying etiologies. Its successful integration in most cases and limited complication rate support its clinical potential. However, given this study’s retrospective design and limited sample size, further prospective studies are required to validate these findings and assess long-term outcomes.

## 1. Introduction

Full-thickness skin defects (FTSDs) still present a challenge in wound management due to the complete loss of both the epidermal and dermal layer [1]. These defects, which can result from full-thickness burns, mechanical trauma, degloving injuries, vascular ulcers, infections, or necrotizing fasciitis, require specialized treatment [2]. Standard of care (SOC) for FTSDs typically involves early debridement to remove non-viable tissues possibly followed by wound bed preparation but ultimately requiring a method of autologous skin reconstruction [1,3,4,5,6,7,8,9,10,11]. For larger FTSDs, autologous split-thickness skin grafting (STSG) with expansion methods like the mesh or Meek (micrografting) technique is usually indicated [12]. Despite their usefulness, STSGs carry a higher risk of hypertrophic scarring and contractures due to the almost complete lack of dermal layer reconstruction [13]. Donor site morbidity is also a concern, particularly in patients with pre-existing conditions like diabetes, vascular disease, or prolonged steroid use, increasing the risk of pain, infection, and delayed wound healing [14,15]. Full-thickness skin grafting (FTSG), consisting of the epidermal and entire dermal layer of the skin, may serve as an alternative for FTSDs. However, since primary closure of the donor site is required, it limits the amount of skin that can be harvested. This technique is therefore indicated when patients have small FTSDs and poor skin quality [16]. The need for complete skin reconstruction has driven the search for alternative solutions, leading to the development of bioengineered and synthetic dermal skin substitutes or dermal regeneration templates (DRTs). DRTs are skin replacements that are placed in the wound bed after sustaining significant damage [17]. DRTs, which can be allogeneic, xenogeneic, or synthetic, provide a scaffold for the formation of a vascularized neo-dermis and facilitate the use of thinner autografts, or ideally even cultured keratinocytes, that minimize donor site morbidity while enhancing aesthetic and functional outcomes [1]. These templates can be used in one- or two-stage procedures depending on whether the covering skin grafts are applied in a simultaneous procedure or delayed procedure, respectively. Although many DRTs have been developed, an ideal, readily available, and affordable dermal matrix has yet to be found.

The NovoSorb^®^ Biodegradable Temporising Matrix (BTM; PolyNovo Biomaterials Pty Ltd., Port Melbourne, VIC, Australia) is a synthetic, bilayered construct, consisting of a layer made up of a polyurethane open-cell foam that interacts with the wound bed and an uppermost second layer, which consists of a non-biodegradable sealing membrane, and can be applied in a two-stage process. In stage one, the BTM is applied onto a prepared wound bed to promote the formation of a vascularized neo-dermis, with the upper sealing membrane protecting the matrix during healing [18]. Wound bed preparation is a critical aspect of wound management and involves various techniques such as mechanical debridement (removal of blisters and necrotic tissue), selective enzymatic debridement [10,19,20,21], or surgical debridement followed by the application of allografts, negative pressure wound therapy (NPWT), etc. These techniques optimize the wound environment and facilitate wound healing [22,23]. We often even use NPWT once the BTM has been applied to enhance the neo-dermis formation. After two to three weeks, a vascularized neo-dermis is formed and capillary refill can be seen [18]. In the second stage, the synthetic sealing membrane that protects the vascularization matrix against infection and dehydration is removed and an STSG is applied when no re-epithelialization occurs with the BTM alone [18]. Alternatively, definitive closure by secondary intention healing can be achieved [18,24,25].

While several clinical publications report satisfactory outcomes with the BTM across various indications [18,26,27,28,29,30,31], the majority of studies originate from Australia, where the product was developed, with additional data from the United States. Only a limited number of studies have been carried out in Europe. We report on our experience with the use of the BTM according to a standardized protocol for various indications in six patients with complex FTSDs. The aim of this study was to evaluate the clinical efficacy of the BTM as a synthetic dermal substitute in the reconstruction of complex FTSDs and to explore its indications, integration success, and outcomes in a European clinical setting.

## 2. Materials and Methods

### 2.1. Patients (Ethics Committee)

From September 2020 to January 2022, the BTM was used for the selective reconstruction of various complex wounds in the Burn Center of Ghent University Hospital. This retrospective case series was approved by the ethical committee of Ghent University Hospital on 9 August 2022 and received the following reference: ONZ-2022-0289. Eligible participants were selected based on the criteria listed in Table 1. Oral and written informed consent for the publication of clinical data and images was obtained from each patient.

### 2.2. BTM Application Protocol for Complex Full-Thickness Wounds


**1st stage: wound bed preparation and BTM application**


The wound bed was surgically, or enzymatically with NexoBrid^®^ (EDNX, MediWound, Yavne, Israel) in the case of full-thickness burns, debrided to remove local debris and dead tissue. Subsequently, signs of critical bacterial colonization or infection were assessed through clinical evaluations and wound swabs. If present, the use of the BTM was delayed until resolved. In the meantime, allografts were applied. Once the wound was adequately prepared, it was re-debrided, measured, and a corresponding piece of BTM was trimmed to fit the wound bed. The foam scaffold side was placed directly onto the wound with the sealing membrane facing upward, and the edges were secured with skin staples. A protective dressing of a Kliniderm^®^ silicone sheet (H&R Healthcare Ltd., Melton, UK), sterile gauze, and an absorbent bandage were applied to prevent dehydration and maintain dressing stability. Debridement and BTM placement were performed by one of the two burn surgeons, SM or KC, both of whom use the same technique.


**Post-surgical care**


Following BTM application, the wounds were assessed daily. According to the SOC protocol, the bandage remained closed on day one post-operation, unless NPWT was applied to enhance the contact between the wound bed and BTM, in which case the bandage was left until the next NPWT dressing change. The use of negative pressure wound therapy was determined on a case-by-case basis. On day two, all dressings were removed and the silicone sheet (Kliniderm^®^) that covered the scaffold was disinfected with povidone-iodine solution (Iso-betadine^®^) (Figure 1A,B). The BTM was gently cleaned with a dry compress to remove any exudate without causing friction (Figure 1C). The wound was thereafter disinfected with povidone-iodine solution using soaked compresses (Figure 1D). The silicone sheet (Kliniderm^®^) was reapplied and the wound was covered with dry sterile compresses and an absorbent bandage (Figure 1E,F). Weekly wound swabs monitored bacterial load, and patients were discharged for weekly follow-ups once clinically stable (absence of clinical signs of infection, appropriate wound bed appearance, absence of pain or systemic symptoms, and continued progression of vascularization and integration of the dermal matrix).


**2nd stage: sealing membrane removal and autografting**


Over time, the scaffold becomes infiltrated by different types of cells, and vascularization of the neo-dermis will occur. Infiltrating cells will use the scaffold as a guide to deposit new extracellular matrix materials, and the tissue will grow into the wound bed. This process usually takes three to four weeks; however, factors such as the size and depth of the wound, its location, the patient’s comorbidities, and the overall status of the patient can lead to longer BTM integration times [2,16,18,32,33,34]. Once adequate vascularization was confirmed—as advised by the manufacturer—by the presence of a salmon-pink color and visible capillary refill, the sealing membrane was removed, the superficial neo-dermis was rubbed gently to check for and improve blood flow, and subsequent final coverage of the defect was achieved using STSGs. The implantation process is illustrated in Figure 2.


**Scar management**


After hospital discharge, all patients were seen weekly in the outpatient clinic until complete wound closure, as confirmed by the plastic surgeon (SM or KC) and documented by photographic imaging. After wound closure, the patients were still regularly seen at the scar clinic at our department for follow-up for customized anti-scar therapy. During this consultation, the patient was seen by two clinical experts (plastic surgeon and physiotherapist) and a certified prosthetist orthothist (CPO). The three pillars of scar treatment consist of hydration with moisturizers and/or silicones, pressure therapy (with or without padding), and UV protection. For optimal effectiveness, the pressure garments should be worn for 23 h per day and should last at least 12 months, ideally extending to 18–24 months [35]. Our previous studies demonstrated the various advantages of these combination therapies [11,35]. During clinical examination, performed by the two scar experts—with more than 30 years of experience in burn and scar treatment—(HH and SM), scars were evaluated. Scars were clinically assessed to determine if they are normal, hypertrophic, or atrophic. The presence of contractures and overall functionality were also evaluated. Scar treatment was adjusted accordingly. At the end of the consultation, digital photographs were taken to monitor the evolution over time, and clinical findings were documented in the medical record of the patient.

## 3. Results

This study included six patients (four male and two female) with an average age of 46.5 years (range: 18–63). The patients presented with various FTSDs including degloving injuries (3), burns (1), ulcerations (1), and necrotizing fasciitis (1). The BTM was selected to treat these cases due to its benefits in minimizing donor site requirements for large defects (patients 1 and 2) and in patients with comorbidities that limit donor site viability (patient 4). Additionally, the use of the BTM helped mitigate hypertrophic scarring and contracture risks, particularly in high-mobility areas like the arm (patients 3 and 5) and hand (patient 6). Most patients had a follow-up period of at least two years, except for two patients who were lost to follow-up at one and three months post-injury. The FTSGs affected various areas, including the upper and lower extremities, as well in the iliac and suprapubic regions. On average, the BTM sealing membrane was removed 20.6 days (range: 13–23) after application, with STSGs (expansion rate range: 1:1–1:2) applied at an average of approximately (in five of the six cases) 39.7 days (range: 15–53) post-injury. Patient demographics, wound characteristics, and outcomes are summarized in Table 2, and individual case descriptions are provided below (Figure 3, Figure 4, Figure 5, Figure 6, Figure 7 and Figure 8).

### 3.1. Case Descriptions

#### 3.1.1. Patient 1

A 40-year-old woman was transferred to our center with persistent sepsis secondary to extensive necrotizing fasciitis after abdominal hernia reduction. Urgent surgical debridement, IV antibiotics, and analgesia were initiated. An FTSD was present at the level of the left iliac region, suprapubic thigh, and proximal thigh, exposing the underlying abdominal and upper leg musculature (Figure 3A). Following three exploratory revisions and further debridement, the wound was treated 25 days after onset with the BTM (Figure 3B) and secured with NPWT for optimal contact. At 22 days post-application, the sealing membrane was removed, revealing a well-perfused neo-dermis (Figure 3C). After gentle abrasion to punctiform bleeding, the integrated matrix was overlaid with the MEEK grafts with an expansion ratio of 1:2, given that the defect measured 1000 cm^2^. One week after removing the plissées, 75% re-epithelialization was obtained. The patient was regularly seen in the outpatient clinic, where our SOC scar management was administered. Nine months post-injury, excellent aesthetic results were observed, with a flexible scar at the left proximal thigh. However, the scar at the left iliac region was very thin and rigid, leading to scar correction with a pedicled flap in order to improve flexibility and restore a better range of motion of the left hip joint. Lateral wound dehiscence occurred, which was treated with povidone iodine and powdered sugar paste. The wound healed by secondary intention, and hydration and pressure therapy were initiated (Figure 3D). Three years post-injury, a favorable aesthetic result was obtained with no hypertrophic scarring or contractures (Figure 3E).

**Figure 3 ebj-06-00024-f003:**
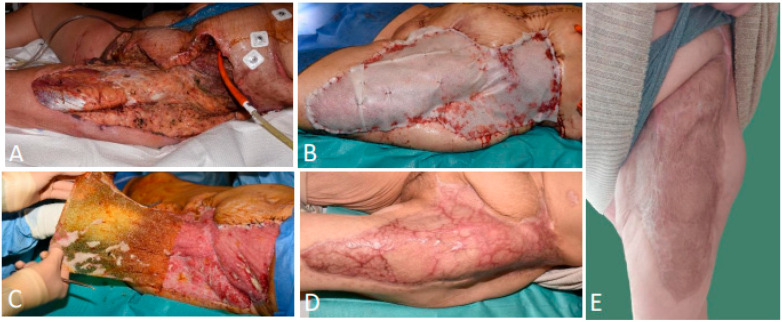
(**A**) An extensive wound defect of the left iliac region, suprapubic thigh, and proximal thigh following necrotizing fasciitis. (**B**) Wound closure with the BTM 25 days after onset. (**C**) Delamination of the sealing membrane 22 days post-application, revealing a well-vascularized neo-dermis. (**D**) An excellent aesthetic result nine months post-injury. (**E**) An excellent aesthetic result three years post-injury.

#### 3.1.2. Patient 2

A 54-year-old man sustained circumferential degloving injuries to his right upper and proximal lower leg after a traffic accident, resulting in sensory loss at the injury site (Figure 4A). Surgical debridement was performed, and the degloved skin was defatted and repositioned as an FTSG, avoiding a donor site. Despite adequate revascularization in 70% of the FTSG, progressive dry necrosis occurred at the medial and lateral aspects of the knee. After debridement of the necrotic tissue on day 16 post-injury, the lacerations were temporarily covered with allografts. One week later, the allografts were removed, revealing a well-vascularized wound bed that was ready to be covered with the BTM (Figure 4B). The patient was in good clinical condition which allowed him to be discharged from the hospital after two days. Clear instructions regarding wound care and limited knee flexion to 120° were also included. The patient was seen weekly in the outpatient clinic. Twenty-three days after application, the BTM was delaminated (Figure 4C). The surface of the neo-dermis was gently abrased to punctiform bleeding before applying the STSG mesh with an expansion ratio of 1:2. Once again, clear post-operative instructions were given, emphasizing this time no knee flexion. Nineteen days after delaminating the BTM, total wound closure was obtained. The patient was regularly seen in the outpatient clinic, where our SOC scar management was administered. Six months after the application of the STSG, small lesions on the lateral aspect of the knee were observed, which were attributed to scar dehydration and friction during mobilization. To promote wound closure and prevent scar contracture in this area, an additional minor STSG was performed. One-and-a-half years post-injury, the aesthetic result was favorable with minimal hypertrophic scarring (Figure 4D). Four years post-injury, the aesthetic result was excellent with no hypertrophic scarring, and the knee’s range of motion was well preserved, allowing for full extension and flexion beyond 90° (Figure 4E).

**Figure 4 ebj-06-00024-f004:**
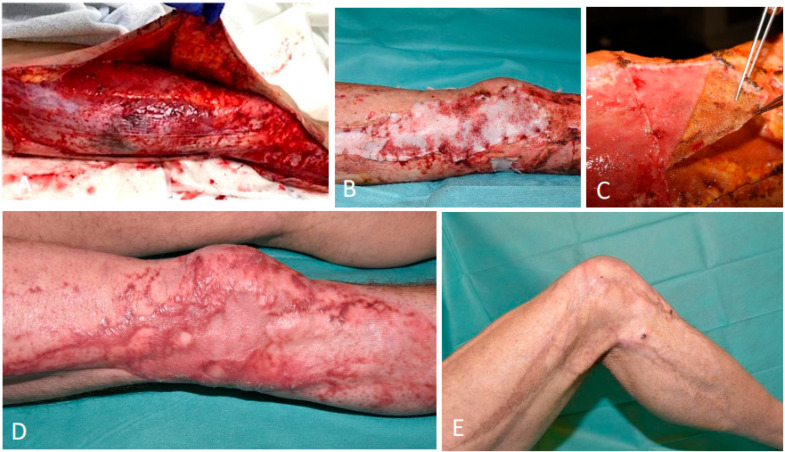
(**A**) Traumatic degloving of the right upper and proximal lower leg. (**B**) Application of the BTM 23 days post-injury to the medial and lateral aspect of the knee. (**C**) Delamination of the sealing membrane, revealing a well-vascularized neo-dermis. (**D**) The right knee 1.5 years after degloving. (**E**) The right knee four years after degloving with favorable aesthetic and functional outcome.

#### 3.1.3. Patient 3

A 17-year-old man was involved in a rollover car crash with his arm dangling out of the window. The patient presented with degloving of the left forearm with underlying tendon ruptures of the palmaris longus and flexor carpi ulnaris muscles (Figure 5A). Initial treatment included debridement, tendon repair, and temporary allograft closure. Four days later, the BTM was applied (Figure 5B) in combination with NPWT and was maintained for one week. The patient was discharged in a good general condition and was seen weekly in the outpatient clinic. Interim evaluations showed progressive integration of the BTM (Figure 5C). Twenty-two days after application, the sealing membrane was removed. The underlying neo-dermis was gently superficially scrubbed to check for and improve blood flow and the STSG mesh with an expansion ratio of 1:1.5 was then applied. After 15 days, 100% graft take was obtained. The patient was regularly seen in the outpatient clinic, where our SOC scar management was administered. This resulted in a favorable aesthetic outcome and flexibility of the skin, with preservation of forearm functionality allowing for full supination and pronation at one year follow-up (Figure 5D). Three years post-injury, the results are excellent with no hypertrophic scarring or functional impairment (Figure 5E).

**Figure 5 ebj-06-00024-f005:**
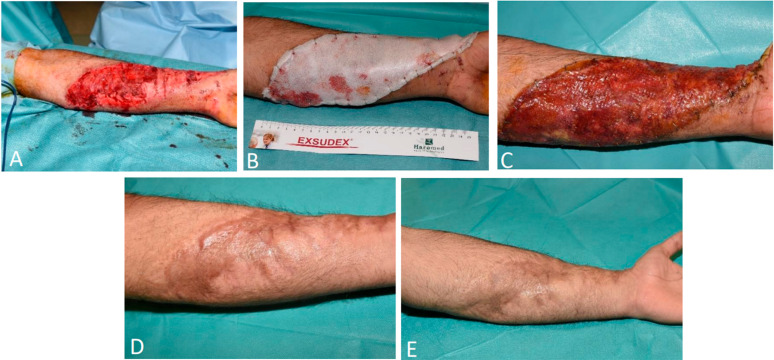
(**A**) Traumatic degloving of the left forearm following a car crash. (**B**) BTM application 4 days post-injury. (**C**) Excellent integration of the BTM to the wound bed 22 days after application. (**D**) STSG outcome one year after degloving. (**E**) Three years post-injury with excellent aesthetic and functional outcome.

#### 3.1.4. Patient 4

A 63-year-old man with peripheral arterial insufficiency and diabetes mellitus presented with chronic arterial ulcerations on his left ankle. Following limited success with percutaneous transluminal angioplasty (PTA) for wound healing, the plastic surgery team was consulted. After surgical debridement of the nonvital tissue, the BTM was stapled to the exposed Achilles tendon and combined with NPWT for a week. At 23 days, after an interim hospital discharge, the sealing membrane was removed, revealing a well-vascularized neo-dermis (Figure 6A), which was covered with STSG mesh with an expansion ratio of 1:1 (Figure 6B). One month later, a yellowish film developed over the wound, beneath which hypergranulation gradually emerged (Figure 6C). A corticosteroid cream was applied to address this, as it could potentially slow down wound healing. At the last check-up, three months after BTM treatment, the wound was less exuding and reduced in size, but was still not completely re-epithelialized due to the underlying vascular problems (Figure 6D). Unfortunately, the patient did not return for follow-up in our department but was treated further in the thoracic and vascular department to optimize his vascular status.

**Figure 6 ebj-06-00024-f006:**
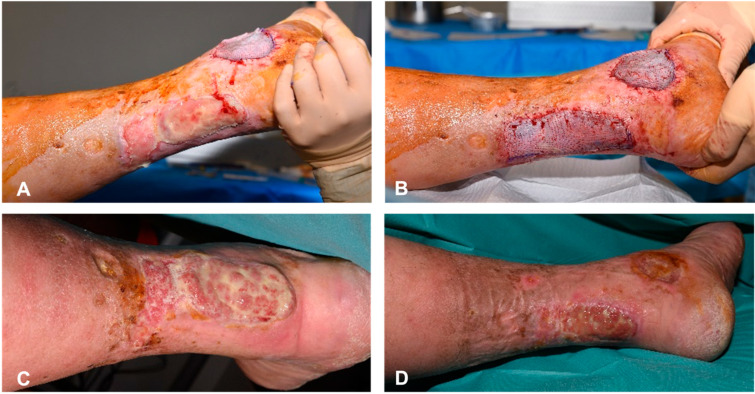
(**A**) The delaminated BTM at the left ankle with a well-perfused neo-dermis, 23 days after application. (**B**) The STSG on the left medial malleolus and Achilles tendon. (**C**) Hypergranulation covered with localized yellow batter, one month after skin graft application. (**D**) Incomplete re-epithelization three months post-application of the BTM with significant wound reduction.

#### 3.1.5. Patient 5

A 63-year-old woman suffered from a degloving injury from her right wrist to axilla in a traffic accident, along with fractures of the ulna and humerus. On the same day, the patient underwent surgical debridement, in which the loose skin was defatted and replaced as an FTSG. After three days, a limited, thin free anterolateral thigh (ALT) flap was harvested and anastomosed at the level of the fossa cubiti. Two weeks later, the necrotizing FTSG was removed because of a fungal infection, which subsequently required two additional debridement revisions (Figure 7A). At 32 days post-injury, the wound was again debrided, dressed with the BTM, and secured with NPWT for optimal contact (Figure 7B). The dermal matrix was placed at the level of the proximal forearm and posterior upper arm, combined with a slight approximation of the free ALT flap. After 21 days, the sealing membrane was removed (Figure 7C), revealing a well-vascularized wound bed, suitable for STSG application with an expansion ratio of 1:2. Sixteen days after delaminating the BTM and STSG application, 100% wound closure was obtained. Two months post-injury, the patient could leave the hospital and was regularly seen in the outpatient clinic, where our SOC scar management was administered. The autograft matured well with no pathological scar development, resulting in a pleasing aesthetic and functional outcome with full extension and 100° flexion range of motion at the 9-month follow-up (Figure 7D). At two-and-a-half years post-injury, no hypertrophic scarring was observed, and functional outcomes remained favorable, with the patient’s range of motion well preserved (Figure 7E,F).

**Figure 7 ebj-06-00024-f007:**
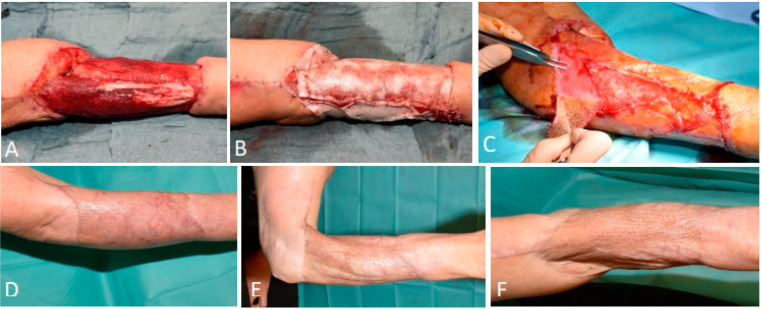
(**A**) Full-thickness skin defect of the right forearm after loss of the necrotizing FTSG. (**B**) Wound dressing with the BTM 32 days post-injury. (**C**) Delamination of the sealing membrane 21 days post-application, revealing a well-perfused neo-dermis. (**D**) Pleasing aesthetic outcome one year post-injury. (**E**,**F**) Excellent aesthetic and functional outcome at two-and-a-half years post-injury.

#### 3.1.6. Patient 6

A 42-year-old man was transferred to our burn unit with contact burns involving the dorsum of the right hand and fingers (TBSA 3.5%). Clinical and laser Doppler imaging (LDI, Moor Instruments, Axminster, UK) confirmed full-thickness burns. The wound was thoroughly cleaned through hydrotherapy and mechanical debridement, and then treated following SOC wound treatment of the burn center consisting of Flaminal Forte (Flen Health, Kontich, Belgium). LDI showed full-thickness burns on day two, which was an indication for enzymatic debridement with EDNX the same day (Figure 8A). After the EDNX procedure, the wound was immediately covered with the BTM (Figure 8B). The clinical presentation was indicative of infection, and the culture was positive for methicillin-susceptible Staphylococcus aureus. The patient was treated with intravenous amoxicillin-clavulanic acid for three weeks, followed by an additional three weeks of oral levofloxacin. Fifteen days after application, partial integration of the BTM was noted (Figure 8C), becoming loose upon sealing membrane removal. This led to extensive necrosis, initially managed with the application of an allograft over the edematous wound bed. One week later, the allograft was replaced with a free serratus anterior fascia flap, covered by an STSG mesh at a 1:1 expansion ratio. This approach was chosen to effectively fill and cover the wound bed while minimizing bulk on the dorsum of the hand—an area where contour and flexibility are particularly important. Two days later, the flap suffered from venous congestion, and extensive venous thromboses were found during revision surgery. Eventually, this led to loss of the free flap and the wound bed was covered with the STSG mesh that initially covered the free flap (Figure 8D). In addition to the local wound care protocol, intravenous and oral antibiotics were administered to the patient until discharge on day 46. The patient was repatriated home and was lost to follow-up.

**Figure 8 ebj-06-00024-f008:**
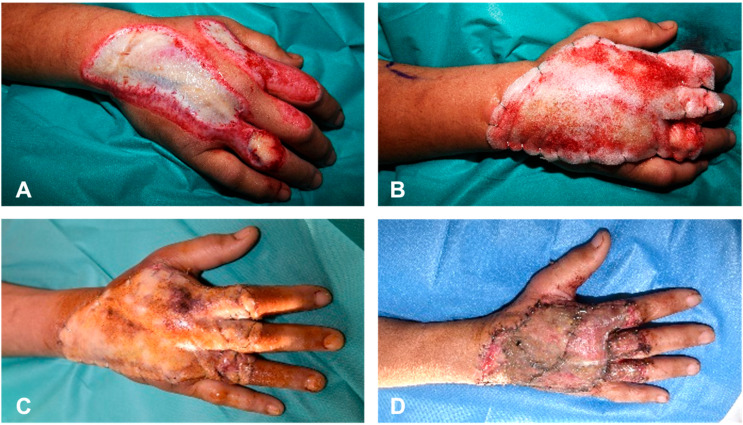
(**A**) A burn at the dorsum of the right hand after enzymatic debridement. (**B**) The BTM stapled to the wound edges. (**C**) A total of 15 days after application showing insufficient integration. (**D**) Revision of STSG application after flap failure one month after the contact burn.

## 4. Discussion

The BTM, a fully synthetic, biodegradable dermal substitute, has gained widespread interest for its effectiveness in managing complex FTSDs due to its versatility and reliability in promoting wound healing and tissue regeneration [2,16,18,24,33,36]. This fully synthetic matrix can convert exposed wound beds into suitable surfaces for skin grafting, rendering it particularly effective in the management of complex FTSDs [2,18]. Although originally designed for burn reconstruction [2,33], the BTM has since been applied to a broader range of wounds, including necrotizing fasciitis, degloving injuries, free flap donor sites, traumatic wounds, and diabetic foot ulcers [2,18,25,31,37,38,39,40,41]. Despite substantial research from centers in Australia and the United States, studies on the BTM from European institutions are limited. In this manuscript, we report on our experience with the use of the BTM in six patients with various FTSDs.

Our findings align with the existing literature regarding the integration timeframe for the BTM [2,18,33]. According to the manufacturer, the BTM requires a successful integration period of two to three weeks [42]. In our study, the average time for BTM integration among the six cases was three weeks. Notably, in four patients, NPWT was applied in addition to the standard attachment of the BTM using skin staples. Recent research conducted by Austin et al. demonstrated that the combination of the BTM and NPWT significantly enhanced integration success [43]. In our series, three patients treated with NPWT achieved successful graft take rates of 75%, 100%, and 100%. One patient, diagnosed with peripheral arterial insufficiency and diabetes mellitus, experienced challenging wound healing and was lost to follow-up, so we do not have the final results for this case. Several studies confirm that the combination of the BTM and NPWT may improve integration rates [44]. Our study observed this trend as well, although we could not demonstrate a significant advantage compared to patients who did not receive NPWT due to small case numbers.

In addition to the BTM, several other dermal substitutes are available on the market, including Integra^®^ Dermal Regeneration Matrix (Integra Life Sciences Corp., Plainsboro, NJ, USA), MatriDerm^®^ (Dr. Otto Suwelack Skin & Health Care AG, Billerbeck, Germany), and Glycerolised Acellular Dermis (Glyaderm^®^, Euro Skin Bank, Beverwijk, The Netherlands). These substitutes provide a biodegradable scaffold that supports wound healing and the regeneration of the neo-dermis [45]. Dermal substitutes can be categorized into single-stage or two-stage applications. Single-stage products are applied directly onto a debrided wound bed and covered immediately with an STSG, while two-stage substitutes are applied in two separate procedures [46]. Integra is a two-stage, bilayered product composed of bovine type I collagen and shark chondroitin-6-sulfate, topped with a removable silicone layer. Although it is widely used for extensive burn reconstruction, it has been associated with higher infection risks and increased treatment costs. Several studies have demonstrated that Integra is more vulnerable to bacterial colonization and requires longer integration times, delaying definitive closure [29,47]. Jou et al. performed a direct cost comparison of the BTM and Integra in upper extremity reconstruction and found that the BTM required fewer secondary procedures and had a significantly lower average cost of USD 1361.92 versus USD 3185.71 for Integra [48]. MatriDerm, by contrast, is a single-stage, monolayer dermal substitute made from bovine type I, III, and V collagen and elastin hydrolysate [49]. Unlike Integra, it is non-crosslinked, allowing for more rapid cellular infiltration, although this also leads to early degradation and a loss of matrix thickness [49]. Glyaderm, a glycerolized, decellularized human dermal matrix, also allows for single-stage application and is significantly thinner (0.4 mm) than commercial synthetic matrices such as the BTM (2.0 mm), Integra (1.5 mm), or MatriDerm (1 mm). Despite its lower thickness, a recent randomized trial has shown improved scar outcomes with Glyaderm in a single-stage protocol, offering a cost-effective alternative due to its low production costs and reduced hospital stay [50]. Compared to these alternatives, the BTM offers the advantage of being fully synthetic and inexpensive to produce, avoiding ethical concerns and risks of disease transmission inherent to biological matrices [29,47]. Furthermore, cost reduction is driven not only by material price but also by fewer complications, fewer surgical interventions, and improved integration outcomes as evidenced by both animal and human studies [25,48].

The BTM offers several advantages over animal or human-derived substitutes (xeno- or allogeneic). First, the BTM is generally considered a less expensive option for the treatment of large, complex FTSDs [47,51]. By reducing the length of hospital stay and the number of required surgical procedures, the BTM can help lower healthcare costs, making it more accessible, particularly for uninsured or underinsured patients [52]. Second, the BTM is readily available and does not always require donor tissue, in comparison to Glyaderm [52,53]. This reduces the risk of donor site morbidity and complications associated with tissue rejection or infection [18,37,47]. Third, its fully synthetic composition eliminates the possibility of disease transmission, offering additional safety benefits and alleviating ethical concerns related to the use of human or animal-derived products [18,29,37]. Finally, the BTM offers a substantial benefit in terms of infection risk reduction [18]. The completely synthetic material contains no elements that can provide bacteria with nutrients [52]. Additionally, the temporary coverage over the wound bed effectively hinders the entry of bacteria and other contaminants into the wound [16,30,31,33,47]. This is particularly advantageous for patients with compromised immune systems, such as burn victims, or those who are at high risk of infections. In this study, we observed two instances of infection. One infection (patient 4) occurred in a high-risk ankle wound due to compromised vascular status and diabetes mellitus. Notably, the BTM has shown strong wound integration even in the presence of infection [25,33,37,40]. Although the infection in patient 4 was observed after STSG application, recent studies report a remarkably low risk of infection in diabetic foot ulcerations following BTM application [25,40]. Kuang et al. [40] reported an infection rate of 15.4% in this high-risk population. Another infection was detected in patient 6 following the application of the BTM. However, a positive wound swab alone does not necessarily indicate a clinical infection, as burn wounds often show signs of contamination or colonization. It is important to distinguish between a clinically significant wound infection—marked by abnormal discharge and/or a foul odor—and a wound that simply has a positive culture. In the case of a clinical infection, the BTM can remain in place, and the wound should be treated topically. Antibiotics should be initiated if necessary. In cases of severe infection, partial or complete removal of the BTM may be required [16,24,32,44]. Overall, infection rates associated with the BTM are lower than with alternative treatments [2,29], likely due to its synthetic composition and rapid vascularization, which enhance its infection resistance [24,29,33,44]. Nevertheless, infection underneath the BTM remains a significant predictor of poor BTM take [24].

The BTM offers both aesthetic and functional benefits. The sealing membrane prevents dehydration, which reduces wound contracture [32]. The scaffold supports generation of a neo-dermis, which leads to less scar tissue formation, and is structured in such a way that it discourages the formation of long stretches of linear collagen, which might lead to wound contracture. In our study, aesthetic outcomes were highly favorable, characterized by flexible skin without hypertrophic scarring or significant contractures, and minimal hyperpigmentation. Although objective scar assessment tools were not used (e.g., Vancouver Scar Scale (VSS) of Patient and Observer Scar Assessment Scale (POSAS) [54,55]), the scars were evaluated by two experienced scar experts (SM and HH), each with over 30 years of clinical expertise. Functionally, outcomes were equally pleasing, with preserved range of motion (notably in the knee of patient 2) and minimal contracture formation. These results align with reports indicating that the BTM contributes to less scar tissue and promotes favorable structural collagen alignment, reducing contracture risk.

Overall, the use of the BTM offers numerous advantages for the reconstruction of large FTSDs. Its ability to reduce infection rates, provide temporary coverage of the wound bed [30,31,33,47], lower healthcare costs [25], eliminate disease transmission risk [18,37,47], and produce aesthetically and functionally pleasing outcomes make it an attractive option for patients and clinicians alike [32]. This case series, one of the few conducted in a European setting, further validates the BTM as an effective, accessible option for complex wound management, with consistent integration and positive patient outcomes, and should therefore be more used across Europe. Its use should be more widely adopted across Europe. As research into this material continues, it is likely that more benefits will be discovered, further increasing its potential in the field of wound healing and tissue reconstruction.

### Limitation

The retrospective design of this study introduces a potential risk of selection bias. Due to its retrospective nature, objective scar assessment using standardized scales was not possible. However, scar evaluations were conducted by two experts (SM and HH), each with over 30 years of experience. Another limitation is the small sample size of only six patients, as well as the varied etiologies of the wounds. These differences limited comparability between cases, although they do highlight the potential versatility of the BTM in treating various wound types. Additionally, two patients were lost to follow-up, making it challenging to fully assess the impact of the BTM on wound healing and scar formation.

## 5. Conclusions

This case series supports the use of the NovoSorb^®^ BTM as a promising treatment option for reconstructing FTSDs, demonstrating its potential to achieve favorable clinical outcomes, including flexible, non-hypertrophic scarring and preserved range of motion—even in anatomically demanding regions and in cases complicated by infection. While standardized scar assessment tools were not employed, scars were evaluated by experienced clinicians with over 30 years of expertise in burn and scar care. Based on their clinical judgment and available photographic documentation, none of the patients developed hypertrophic scarring. Nevertheless, we recognize that future studies would benefit from the inclusion of validated objective scar scales to strengthen the reproducibility and comparability of outcomes. Importantly, the BTM’s synthetic and biocompatible design minimizes infection risk, provides robust temporary wound coverage, avoids the need for donor dermis, and can contribute to reduced healthcare costs. These attributes make it a valuable alternative for both patients and clinicians managing complex FTSDs, particularly in high-risk individuals or when donor site availability is limited. The successful application of the BTM in our series benefited from adherence to a standardized protocol adapted to our center’s context and from surgical expertise in burn and reconstructive techniques. While our findings are encouraging, prospective studies with larger cohorts and standardized outcome measures are warranted to confirm long-term efficacy, expand indications, and compare its performance and cost-effectiveness to other dermal substitutes.

## Figures and Tables

**Figure 1 ebj-06-00024-f001:**
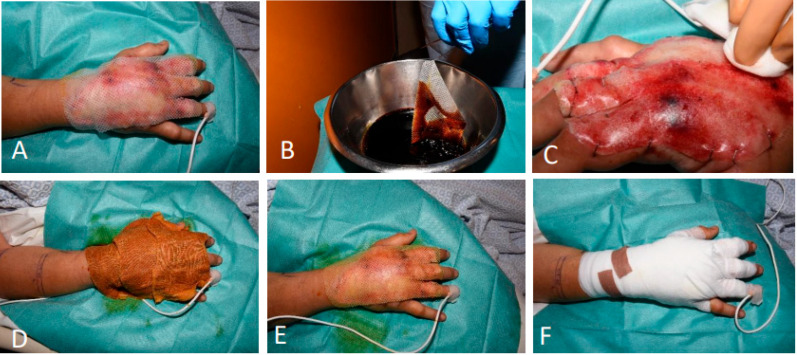
(**A**) Bandage removal with presence of silicone sheet (Kliniderm^®^). (**B**) Disinfection of silicone sheet (Kliniderm^®^) with povidone-iodine solution. (**C**) Gentle cleaning of BTM with dry compress. (**D**) Disinfection of wound using povidone-iodine solution-soaked compresses. (**E**) Reapplication of disinfected silicone sheet (Kliniderm^®^). (**F**) Covering of wound with sterile compresses and absorbent bandage.

**Figure 2 ebj-06-00024-f002:**
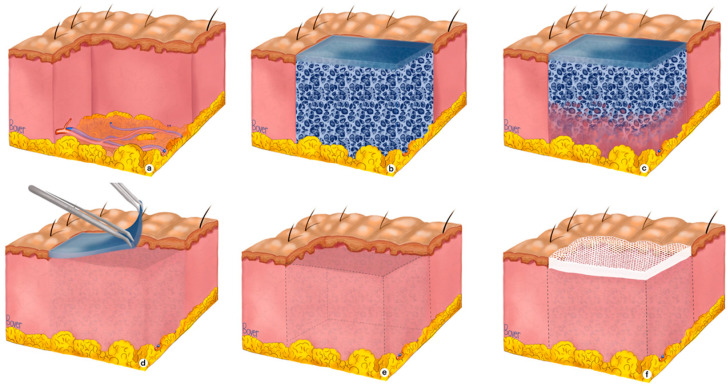
The mechanism of action of the BTM. (**a**) A full-thickness skin defect with complete destruction of both epidermal and dermal layers. (**b**) Reconstruction of the soft-tissue defect by applying the BTM onto the wound bed after proper wound bed preparation and fixating the matrix with staples or sutures. Temporary wound coverage prevents dehydration, temperature shifts, and minimizes bacterial contamination. (**c**) Budding capillaries enter the synthetic matrix and vascularize the avascular layers. (**d**) After complete vascularization of the matrix, the protective sealing membrane is peeled off relatively easily with forceps. (**e**) The neo-dermal layer is exposed and prepared for final coverage. (**f**) Definitive wound closure can be achieved by placement of a skin graft on a well-vascularized neo-dermis.

**Table 1 ebj-06-00024-t001:** Eligibility criteria.

Inclusion Criteria	Exclusion Criteria
Adult patients with full-thickness skin defects Able to provide written informed consent	Children and adolescents (<18 y) Patients with superficial and partial thickness skin defects Inability to provide informed consent due to health condition or refusal to sign

**Table 2 ebj-06-00024-t002:** Patient demographics and wound characteristics. Y, years; F, female; M, male; d, days; BTM, biodegradable temporising matrix; STSG, split-thickness skin graft; NPWT, negative pressure wound therapy.

	Patient Demographics		Wound Characteristics						
Case	Age (y)	Gender	Localization	Etiology	Time to BTM (d)	Time to STSG (d)	Total BTM Integration Time (d)	Complications	NPWT After BTM Application
1	40	F	Upper leg Suprapubic Iliac region	Necrotizing fasciitis	25	47	22	None	Yes
2	54	M	Leg	Degloving	23	46	23	None	No
3	18	M	Forearm	Degloving	4	26	22	None	Yes
4	63	M	Ankle	Ulcerations	28	51	23	Hypergranulation with secondary infection	Yes
5	63	F	Forearm	Degloving	32	53	21	None	Yes
6	42	M	Hand	Burn	2	15	13	Partial integration	No

## Data Availability

The data are contained within the article.

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
