# Peer review of "The Reconstruction of Various Complex Full-Thickness Skin Defects with a Biodegradable Temporising Matrix: A Case Series"

_2673-1991, 2025, doi:10.3390/ebj6020024_

Round 1

Reviewer 1 Report

Comments and Suggestions for Authors

Dear authors,

Thank you for considering EBJ for the publication of your article, "Reconstruction of various complex full-thickness skin defects  with a biodegradable temporising matrix: a case series".

General Comments:

This is an exceptionally well-written article that effectively addresses a critical topic. The main question answered by your research is addressing the use of NovoSorb® Biodegradable Temporizing Matrix (BTM) for full-thickness skin defect (FTSD) reconstruction. The research is relevant, timely, and contributes to the growing body of evidence on synthetic dermal substitutes.

The topic is relevant, and the case series are well presented and each patient's description has clearly formulated background which is also supported by robust evidence from the literature. The topic is original and makes a significant contribution to the field by bridging the current gap in knowledge. Your article offers valuable insights into this area of clinical research. It highlights the advantages of BTM over traditional techniques, particularly its role in minimizing donor site morbidity. The standardized protocol for BTM application enhances the reproducibility of the findings. The diversity of cases (burns, degloving injuries, ulcerations, necrotizing fasciitis) demonstrates the versatility of BTM.

You might consider using objective scar assessment tools, such as the Vancouver Scar Scale (VSS) or Patient and Observer Scar Assessment Scale (POSAS), to quantify scar quality. You might also consider basic cost breakdown, comparing BTM to standard of care (SSG). Should direct cost data is unavailable, a qualitative discussion on potential cost savings (e.g., fewer procedures, reduced hospital stay) would add credibility.

From our perspective, minor improvements are necessary. The conclusions are well-founded and comprehensively address the research question. The references are appropriate, and additional tables or figures are not required to enhance the manuscript.

We have no further comments.

Reviewer 2 Report

Comments and Suggestions for Authors

I want to begin by acknowledging the excellent work in preparing this document. The quality of the information presented is remarkable and provides a clear overview of the topic. However, to further strengthen the findings, I believe it would be appropriate to address some questions that could enrich the analysis. The following are the questions I consider important for a deeper and more complete understanding of the content:

Introduction Section:
1. Are there meta-analyses or systematic reviews that provide an overview of alternatives to autologous grafts, and what are the comparative advantages and disadvantages of allogeneic, xenogeneic, and synthetic DRTs?
2. What criteria define an ideal dermal matrix, and what are the mechanisms of action of NovoSorb® BTM compared to other DRTs?

Materials and Methods Section:
3. What are the limitations of the retrospective case series design regarding bias and generalizability, and what criteria were used to assess adequate vascularization before the second stage?
4. Could you explain the reasons for ruling out negative pressure therapy in all cases and what clinical stability means at weekly follow-up visits?

Results Section:
5. What are the implications of the two patients' failure to follow up for interpreting the results, and could you provide a table or graph comparing the patients' healing times?

Patient 1:
6. Could you detail the "hydration and pressure therapy" protocol used in scar management and explain why the pedicle flap was performed and its advantages over other techniques?

Patient 2:
7. Could you please describe the "small lesions" on the lateral aspect of the knee more precisely, justify the need for an additional STSG, and explain the indications for wound care and the limited knee flexion the patient was given?

Patient 3:
8. Could you please provide more details about the "SOC treatment" used in the outpatient follow-up and explain the type of tendon repairs performed?

Patient 4:
9. Please explain the "yellowish film" and "hypergranulation" observed, their clinical implications, and why the corticosteroid cream is applied. Also, the implications of loss to follow-up in this patient, considering his comorbid condition, will be discussed.

Patient 5:
10. Could you justify using a thin, limited ALT flap instead of other reconstruction options and explain why the serratus anterior fascia free flap was chosen?

Patient 6:
11. Could you provide a detailed description of the Staphylococcus aureus infection, the antibiotics administered, and their duration? Also, considering the complexity of this patient's case, could you discuss the implications of a loss to follow-up and explain why the serratus anterior fascia-free flap was performed?

Discussion Section:
12. Acknowledging the study's limitations, such as the small sample size and the heterogeneity of the wounds treated, is essential, as these variables may influence the generalizability of the results and their applicability in different clinical settings. Furthermore, a more precise explanation of the differentiation between colonization and infection is required, as this distinction is crucial for interpreting the results and understanding the true impact of treatment.

13. Cost and outcome assessment in clinical practice is essential to optimize the use of BTM in healthcare. A thorough analysis of associated costs, comparing them with therapeutic alternatives, is essential to determine the economic viability of the treatment. Furthermore, including quantitative data on aesthetic and functional outcomes, rather than relying solely on qualitative descriptions, will allow for a more objective assessment of treatment efficacy. Finally, discussing the clinical implications of these findings in the European context will help contextualize the results and provide practical recommendations for their implementation in clinical practice.

14. Evidence supporting the claim that BTM is more cost-effective than other dermal substitutes is needed to justify its use and encourage its adoption in clinical practice. Furthermore, the concept of topical treatment in managing wound infections treated with BTM needs to be clarified, which will help improve the understanding of treatment strategies and their effectiveness.

Conclusions Section:
15. Please avoid superlative language such as "highly effective" and "consistent results," and emphasize the need for prospective, controlled studies to validate your findings. Also, clarify the meaning of "standardized protocol" and detail what a "surgeon with extensive experience in dermal replacement techniques" entails.

And restructure the abstract based on the requested modifications.

Reviewer 3 Report

Comments and Suggestions for Authors

Comments for Authors

The manuscript under review “Reconstruction of various complex full-thickness skin defects 2 with a biodegradable temporising matrix: a case series” presents valuable research into the development of materials for wound healing through the presentation of 6 case reports. However, there are areas where the clarity, structure and depth of discussion can be improved. The comments below tend to improve the quality of the manuscript.

Minor Comments

1- The objectives of the study should be more clearly defined in the abstract and introduction. It is not clear from the manuscript whether the aim is to test the efficacy of polyurethane dermal substitute or to test the efficacy of a new protocol combining the application of BTM followed by autologous skin grafting.

2- Provide further justification for the choice of materials and methods. A comparison with previously reported results or with other currently used treatments is recommended.

3- Acronyms such as STSG "autologous split-thickness skin graft" have been defined in the text after their introduction. Please define all acronyms used for the first time and continue to use them in the text.

4- Expand the discussion of the implications of the results. Some comparisons with previous studies could be developed in depth which would strengthen the discussion.

5- Some sentences are too complex. Simplify to improve readability.

6- Reference letters in Figure 2 are absent, include them for clarification.

This study adds valuable knowledge to the field of wound healing materials and provides evidence of an effective protocol for tissue reconstruction. Incorporating the above comments will greatly improve the manuscript and increase its impact. I recommend its publication in the European Burn Journal with Minor Revisions.

Comments on the Quality of English Language

Some sentences are too complex. Simplify to improve readability.

Reviewer 4 Report

Comments and Suggestions for Authors

Dear Editor

Thank you for the opportunity to review the following manuscript “Reconstruction of various complex full-thickness skin defects with a biodegradable temporising matrix: a case series for consideration for the European Burns journal.  This is a well written article however there is a lack of detail regarding outcome measures and scar assessments/ interventions.

Abstract

This section will require minor revisions once changes are made to the results and discussion.   

Introduction

Very well written.

Materials and Methods

Please address recruitment and selection bias.  

2.2. please clarify if all 1st stage procedures were completed by the same surgeon to ensure the same technique was followed

Lines 140-142 references required

Scar management – this section is lacking detail. Please provide details on type of garment, class of compression, type of hydration, type of silicone – wearing regime, frequency of massage, frequency of therapy, home range of motion program.

Please clarify what scar assessments were completed

Please clarify what formal outcome measures were used to measure function?

Results

Line 221 – a scar remains – it is not active – please rephrase this sentence.

Line 238 please clarify the exact knee range of motion limitations e.g. can flex to 30 deg

Please outline scar management interventions for patient two.

Line 247 scar remains – it is not active – please rephrase this sentence.

Please outline scar management interventions for patient three.

How was function measured for patient three?

Linne 279 scar remains – it is not active – please rephrase this sentence.

Please outline scar management interventions for patient five.

How was function and range of motion measured for patient five?

Linne 323 scar remains – it is not active – please rephrase this sentence.

Line 338 please include the waiting time for nexobrid and this would be impacting time to surgery

Please clarify if patient six received any scar management interventions

Discussion

Line 384 – please discuss the costs of two surgeries and how this could impact rather than a single stay procedure with an alternative dermal substitute.

Lines 418-422 this section requires revision. There is a lack of detail in the results above to support this section.

Lines 426-428 this is not supported without measurement – please revised.

Conclusion

This section will require minor revisions once changes are made to the results and discussion.   

Round 2

Reviewer 2 Report

Comments and Suggestions for Authors

no comments

Author Response

Thank you.

Reviewer 3 Report

Comments and Suggestions for Authors

In my opinion, the review carried out by the authors has significantly improved the quality of the article, providing important details regarding the aims, scar treatment, comparisons with other dermal matrices and limitations.

For this reason, I recommend its publication in european burn journal.

Author Response

Thank you.

Reviewer 4 Report

Comments and Suggestions for Authors

Thank you for revising your manuscript however the discussion requires further work. Please appropriately address the feedback regarding the costs of two surgeries and how this could impact rather than a single stay procedure with an alternative dermal substitute within the manuscript. 

Lines 478 -486 have not been adequately addressed - only a limitation has been addressed. As no formal outcome measures were used this cannot be stated. Please respond appropriately to the feedback and address this in the manuscript.  Subjective assessment are not outcomes. 
